# Structural aging of human neurons is opposite of the changes in schizophrenia

**Ryuta Mizutani**[1]*, **Rino Saiga**[1], **Yoshiro Yamamoto**[2], **Masayuki Uesugi**[3], **Akihisa Takeuchi**[3], **Kentaro Uesugi**[3], **Yasuko Terada**[3], **Yoshio Suzuki**[4], **Vincent De Andrade**[5], **Francesco De Carlo**[5], **Susumu Takekoshi**[6], **Chie Inomoto**[7], **Naoya Nakamura**[7], **Youta Torii**[8], **Itaru Kushima**[8,9], **Shuji Iritani**[8,10], **Norio Ozaki**[8¤], **Kenichi Oshima**[10,11], **Masanari Itokawa**[10,11], **Makoto Arai**[11]

**1** Department of Bioengineering, Tokai University, Hiratsuka, Kanagawa, Japan, **2** Department of Mathematics, Tokai University, Hiratsuka, Kanagawa, Japan, **3** Japan Synchrotron Radiation Research Institute (JASRI/SPring-8), Sayo, Hyogo, Japan, **4** Photon Factory, High Energy Accelerator Research Organization KEK, Tsukuba, Ibaraki, Japan, **5** Advanced Photon Source, Argonne National Laboratory, Lemont, IL, United States of America, **6** Department of Cell Biology, Tokai University School of Medicine, Isehara, Kanagawa, Japan, **7** Department of Pathology, Tokai University School of Medicine, Isehara, Kanagawa, Japan, **8** Department of Psychiatry, Nagoya University Graduate School of Medicine, Nagoya, Aichi, Japan, **9** Medical Genomics Center, Nagoya University Hospital, Nagoya, Aichi, Japan, **10** Tokyo Metropolitan Matsuzawa Hospital, Setagaya, Tokyo, Japan, **11** Tokyo Metropolitan Institute of Medical Science, Setagaya, Tokyo, Japan

¤ Current address: Pathophysiology of Mental Disorder, Nagoya University Graduate School of Medicine, Nagoya, Aichi, Japan
* mizutanilaboratory@gmail.com

**Data Availability Statement:** All relevant data are within the paper and its Supporting Information files.

**Funding:** This research was supported by the Japan Agency for Medical Research and

## Abstract

Human mentality develops with age and is altered in psychiatric disorders, though their underlying mechanism is unknown. In this study, we analyzed nanometer-scale three-dimensional structures of brain tissues of the anterior cingulate cortex from eight schizophrenia and eight control cases. The distribution profiles of neurite curvature of the control cases showed a trend depending on their age, resulting in an age-correlated decrease in the standard deviation of neurite curvature (Pearson's $r = -0.80$, $p = 0.018$). In contrast to the control cases, the schizophrenia cases deviate upward from this correlation, exhibiting a 60% higher neurite curvature compared with the controls ($p = 7.8 \times 10^{-4}$). The neurite curvature also showed a correlation with a hallucination score (Pearson's $r = 0.80$, $p = 1.8 \times 10^{-4}$), indicating that neurite structure is relevant to brain function. This report is based on our 3D analysis of human brain tissues over a decade and is unprecedented in terms of the number of cases. We suggest that neurite curvature plays a pivotal role in brain aging and can be used as a hallmark to exploit a novel treatment of schizophrenia.

## Introduction

Human mentality changes with age and matures even through adulthood. This depends on a wide variety of factors, such as genetic background, education, and lifestyle [1]. These factors in life course should affect the brain, including brain areas that exert mental functions. The volumetric decrease of grey matter gradually progresses with age [2, 3], indicating that some

Development https://www.amed.go.jp/ (JP21wm0425007 and JP21dk0307103 to NO, JP19km0405216 and JP22tm0424222 to IK, and JP22wm0425019 to Y Torii and SI) and by the Japan Society for the Promotion of Science https://www.jsps.go.jp/ (20K20602 and 21H04815 to NO, and 21K07543 and 21H00194 to IK). The funders had no role in study design, data collection and analysis, decision to publish, or preparation of the manuscript.

**Competing interests:** I have read the journal's policy and the authors of this manuscript have the following competing interests: Masanari Itokawa and Makoto Arai declare a competing interest, being authors of patents regarding therapeutic use of pyridoxamine for schizophrenia. The patent title is "Detection and treatment of schizophrenia" (US-2014335517-A1, JP 5288365, and EP 2189537). This does not alter our adherence to PLOS ONE policies on sharing data and materials. All other authors declare no competing interest.

microscopic changes occur in human brain tissue. Since loss of cortical neurons is not predominant in normal aging [4–8], the decrease in grey matter volume should be attributed to other factors, such as morphological alteration of neurons [7, 8]. Indeed, microscopic studies of human cortical neurons have revealed significant regression of dendritic arbors with age [7, 9, 10]. However, age-related neuronal alteration and its relation to mental function are not fully understood.

Schizophrenia is a mental disorder showing psychiatric symptoms including hallucinations, delusions and social withdrawal. Since schizophrenia onset peaks at young adult ages [11, 12], brain development and aging should be relevant to the etiology of schizophrenia. Macroscopic brain changes, such as ventricular dilation, have been repeatedly observed in schizophrenia [13–18]. Although the macroscopic changes in the brain should originate from microscopic alterations in the brain tissue, the neuropathology of schizophrenia remains unclarified [19–21].

We recently reported nanometer-scale three-dimensional analyses of post-mortem brain tissues of schizophrenia and control cases [22, 23]. The results of the analysis indicated that neurite curvature differs between individuals and becomes extraordinary in schizophrenia. In this study, we further analyzed brain tissues of the Brodmann area 24 (BA24) of the anterior cingulate cortex of schizophrenia and control cases by using synchrotron radiation nanotomography (nano-CT). In conjunction with our previous results, the geometric analysis of the neuronal structures of eight schizophrenia and eight control cases was performed to identify statistically-significant differences between the schizophrenia and control groups. The neuronal structure was also examined by taking account of age and a hallucination score in order to reveal possible relations between the neuronal structure and clinical information.

## Methods

### Cerebral tissues

All post-mortem human cerebral tissues were collected with written informed consent from the legal next of kin using protocols approved by the Clinical Study Reviewing Board of Tokai University School of Medicine (application no. 07R-018) and the Ethics Committee of Tokyo Metropolitan Institute of Medical Science (approval no. 20–19), as reported previously [22–24]. This study was conducted according to the Declaration of Helsinki under the approval of the Ethics Committee for the Human Subject Study of Tokai University (approval nos. 22009 and 22010), as reported previously [22–24]. The number of cases used for the analysis was determined from the available beamtime at synchrotron radiation facilities. Schizophrenia patients S1–S4 and controls N1–N4 are the same as in our previous study [22]. Schizophrenia patients S1–S8 (S1 and S2 Tables) were diagnosed according to the DSM-IV criteria by at least two experienced psychiatrists. The S8 case carried a chromosome 22q11.2 deletion [25]. Auditory hallucination scores of the S1–S8 cases were estimated with the Present State Examination (PSE) [26] without structural analysis information. Control patients N1–N8 (S1 and S2 Tables) were hospitalized due to traffic injury (N1), sudden death (N7) or non-psychiatric diseases (N2–N6, and N8) and were not psychiatrically evaluated. No records of schizophrenia were found for the control cases. Auditory hallucination scores of the control cases were estimated to be zero [27].

Tissues of the anterior cingulate cortex (Brodmann area 24) were dissected from the left hemispheres of the autopsied brains of the schizophrenia and control cases. Histological assessment of the cerebral tissues showed no hemorrhage, infarction, or neoplasm. The tissues were subjected to Golgi impregnation and embedded in epoxy resin using borosilicate glass capillaries, as reported previously [22].

## Microtomography and nanotomography

The overall structure of the resin-embed samples was visualized with simple projection microtomography at the BL20XU beamline [28] of SPring-8. The data collection conditions are summarized in S3 Table. Absorption contrast images were acquired using CMOS imaging detectors (ORCA-Flash2.8 and ORCA-Flash4.0, Hamamatsu Photonics, Japan) as reported previously [22]. Three-dimensional images of the tissue samples were reconstructed from the obtained x-ray images and used to locate layer V positions to find neurons.

Each neuron was then visualized with synchrotron radiation nanotomography (nano-CT) equipped with Fresnel zone plate optics [29]. The nano-CT experiments were conducted as reported previously [22, 23] at the BL37XU [30] and the BL47XU [31] beamlines of the SPring-8 synchrotron radiation facility, and at the 32-ID beamline [32, 33] of the Advanced Photon Source (APS) of Argonne National Laboratory. The data collection conditions are summarized in S3 Table. Photon flux at the sample position was determined to be $1.4 \times 10^{14}$ photons/mm$^2$/s at BL37XU or $1.5 \times 10^{14}$ photons/mm$^2$/s at 32-ID by using Al$_2$O$_3$:C dosimeters (Nagase-Landauer, Japan). Spatial resolutions were estimated from the Fourier domain plot [34] or by using three-dimensional test patterns [35]. The date range in which the data were collected is from Dec 2011 to Jun 2021. The following structural analysis was conducted from Dec 2011 to Sep 2022.

## Structural analysis

Tomographic reconstruction of the obtained datasets (S2 and S4 Tables) were performed with the convolution-back-projection method using the RecView software (RRID: SCR_016531; https://mizutanilab.github.io/) [36], as reported previously [22]. The reconstruction calculation was conducted by RS. The resultant image datasets were provided to RM without case information in order to eliminate human biases in the structural analysis. RM built three-dimensional coordinate models of tissue structures by tracing the reconstructed images with the MCTrace software (RRID: SCR_016532; https://mizutanilab.github.io/) [37]. Coordinate files of the resultant structural models in Protein Data Bank format were locked down after the model building of each dataset was finished. Then, RM reported the number of neurite segments to RS only by stating that the number should be aggregated into each case. RS aggregated the numbers in order to find cases in which the analysis amounts were less than others. The aggregation results were not returned to RM. Datasets to be further analyzed were chosen by RS according to the aggregation results and provided to RM without case information. These model-building and result aggregation processes were repeated for five batches. Since the first batch consisting of datasets S7A (dataset A of schizophrenia case S7), S7B, S7C, N6F, N6G, N6H, N6I, N7E, N7F, N7G, and N7H (S4C, S4F and S4G Table) was analyzed at the early stage of this study, their case information was disclosed to RM after the model building of the first batch was finished in order to examine the feasibility of the analysis process. The other four batches were continuously analyzed without case information. After the model building of the four batches was finished, all case information was disclosed to RM to assign datasets to each case.

Since eight dummy datasets unrelated to this study were included in order to shuffle datasets, those dummy sets were not used in the subsequent analysis. All other 54 datasets were subjected to geometric analysis. Structural parameters were calculated from three-dimensional Cartesian coordinates by using the MCTrace software, as reported previously [22]. Statistics of the obtained structural parameters are summarized in S1, S2, and S4 Tables.

## Statistical tests

Statistical tests of the structural parameters were performed using the R software, as reported previously [22–24]. Significance was defined as $p < 0.05$. Differences in means of structural

parameters between groups were examined using two-sided Welch's *t*-test. Since Shapiro-Wilk normality tests of standard deviations of the neurite curvature showed no statistical significances (*p* = 0.30 for schizophrenia cases and *p* = 0.94 for controls), the deviation difference between the groups was examined using two-sided Welch's *t*-test. Correlations between structural parameters and clinical information were examined by using linear regression analysis along with Pearson's correlation coefficient.

## Results

### Three-dimensional visualization and structural analysis

A three-dimensional visualization of brain tissues of four schizophrenia (S5–S8) and four control cases (N5–N8; S1 Table) was performed by using synchrotron radiation nano-CT (S3 Table), as previously reported for four other schizophrenia (S1–S4) and four control cases (N1–N4; S2 Table) [22]. The obtained three-dimensional images of neurons were traced to build Cartesian coordinate models of neurons [37] (S1–S3 Figs). Examples of overall images of the tissue samples are shown in S4 Fig. Fig 1 shows a schematic representation of the nano-CT analysis. The figure also shows a three-dimensional image of a neuron of the S8 case carrying a chromosome 22q11.2 deletion [25] and its model built by tracing the image.

The obtained coordinate models are composed of approximately 140 mm of neurite traces in total (S1 Table). Their three-dimensional coordinates were used for calculating the

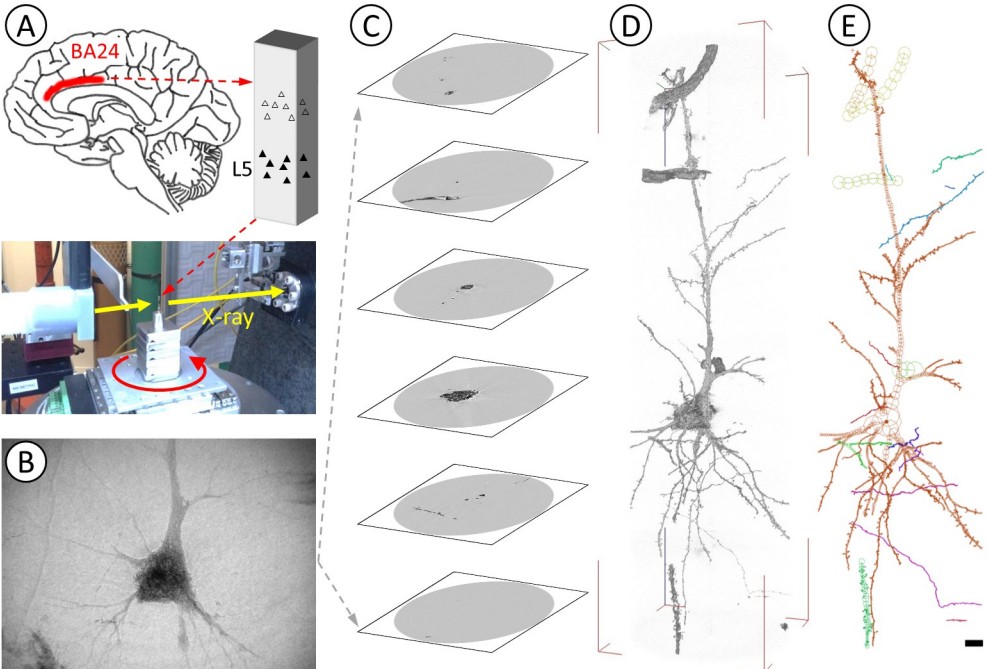

**Fig 1. Nano-CT analysis of human brain neurons.** (**A**) Brain tissues of Brodmann area 24 (BA24) of the anterior cingulate cortex were stained using the Golgi method and mounted on the sample stage of synchrotron radiation nano-CT. (**B**) A series of X-ray images of layer V neurons were taken while rotating the sample. This data collection process was repeated by shifting the sample along the sample rotation axis so as to cover the neuronal arborization. (**C**) Tomographic slices were reconstructed from the image series and then stacked to obtain a three-dimensional image of the neuron. (**D**) Three-dimensional rendering of dataset S8A of the schizophrenia S8 case (S4 Table). Voxel values 60–800 were rendered with the scatter HQ algorithm of the VG Studio Max software (Volume Graphics, Germany). (**E**) The three-dimensional image was traced to build a Cartesian coordinate model by using the MCTrace software [37]. Geometric parameters were calculated from the resultant three-dimensional coordinates. Structural constituents are color-coded. Nodes composing the model are depicted as circles. Scale bar: 10 μm.

geometric parameters of the neuronal network, such as curvature, which is the reciprocal of the curve radius. Statistics of the geometric parameters are summarized in S1, S2 and S4 Tables. Individual curvature data are provided in S1 Data. These are based on our 3D analysis of human brain tissues over a decade and is unprecedented in terms of the number of cases.

## Structure of control cases

The analysis of the obtained geometric parameters along with our previous results [22] indicated that the distribution of neurite curvatures of the control cases varies depending on their age. Fig 2A shows bee-swarm plots of the neurite curvature of the control cases. The plots illustrate that (1) the curvature maximum decreases as age increases and that (2) the curvature median increases as age increases. The decrease in the maximum curvature leads to an age-dependent decrease in the distribution span (Fig 2A), giving rise to a decrease in the standard deviation by age (Fig 2B). This age-dependent distribution resulted in a significant correlation between age and the standard deviation of neurite curvature (Fig 2B; Pearson's $r$ = -0.80, $p$ = 0.018). Since schizophrenia cases deviate upward from the correlation (Fig 2B), the age-related change in the control cases is opposite to the alteration in schizophrenia.

The bee-swarm plots of the neurite curvature of the control cases (Fig 2A) also illustrated that the distribution profiles of the control cases in the 40s exhibit vertically-symmetric spindle shapes, where the profiles of the older cases differ between individuals and hence are unique to each case. This suggests that the neurites of the control cases underwent structural changes during aging in different ways for each case.

## Structural differences between schizophrenia and control groups

The nano-CT analysis also revealed significant differences in the structural parameters between the schizophrenia and control groups. The neurite curvature of the schizophrenia group is significantly higher than that of the control group (Fig 3A; $p$ = 7.8 × 10$^{-4}$, Welch's $t$-test), resulting in a group mean (0.58 μm$^{-1}$) 60% higher than that of the controls (0.36 μm$^{-1}$). As observed in the age-dependence plot (Fig 2B), the standard deviation of neurite curvature is significantly higher in the schizophrenia cases compared with the controls ($p$ = 3.8 × 10$^{-4}$, Welch's $t$-test). The neurites of the schizophrenia cases are significantly thinner than those of the controls (Fig 3B; $p$ = 0.028, Welch's $t$-test). The neurite thinning and curvature increase in schizophrenia cases coincides with our previous observation that neurite thickness inversely correlates with neurite curvature [22, 23].

These structural differences between the schizophrenia and control groups originated from differences in the distribution profiles of neurite curvature. Fig 4A shows the frequency distribution of the neurite curvatures of the schizophrenia cases. The plots exhibit long tails representing the tortuous and thin network of schizophrenia (Fig 4C). In contrast, the control cases showed almost no distribution beyond 0.8 μm$^{-1}$ in neurite curvature (Fig 4B), representing a rather smooth and thick network (Fig 4D). The parameter differences between the schizophrenia and control groups (Figs 2 and 3) are due to these structural differences.

## Neurite curvature correlates with hallucination score

Fig 5 shows a scatter plot of neurite curvature and auditory hallucination score. The plot indicated a significant correlation (Pearson's $r$ = 0.80, $p$ = 1.8 × 10$^{-4}$), showing that the neuron structure correlates with the psychiatric symptom. We suggest that the hallucination severity can be estimated from a structural analysis of brain tissue by using this plot. No obvious correlation was found between the structural parameters and chlorpromazine equivalent dose

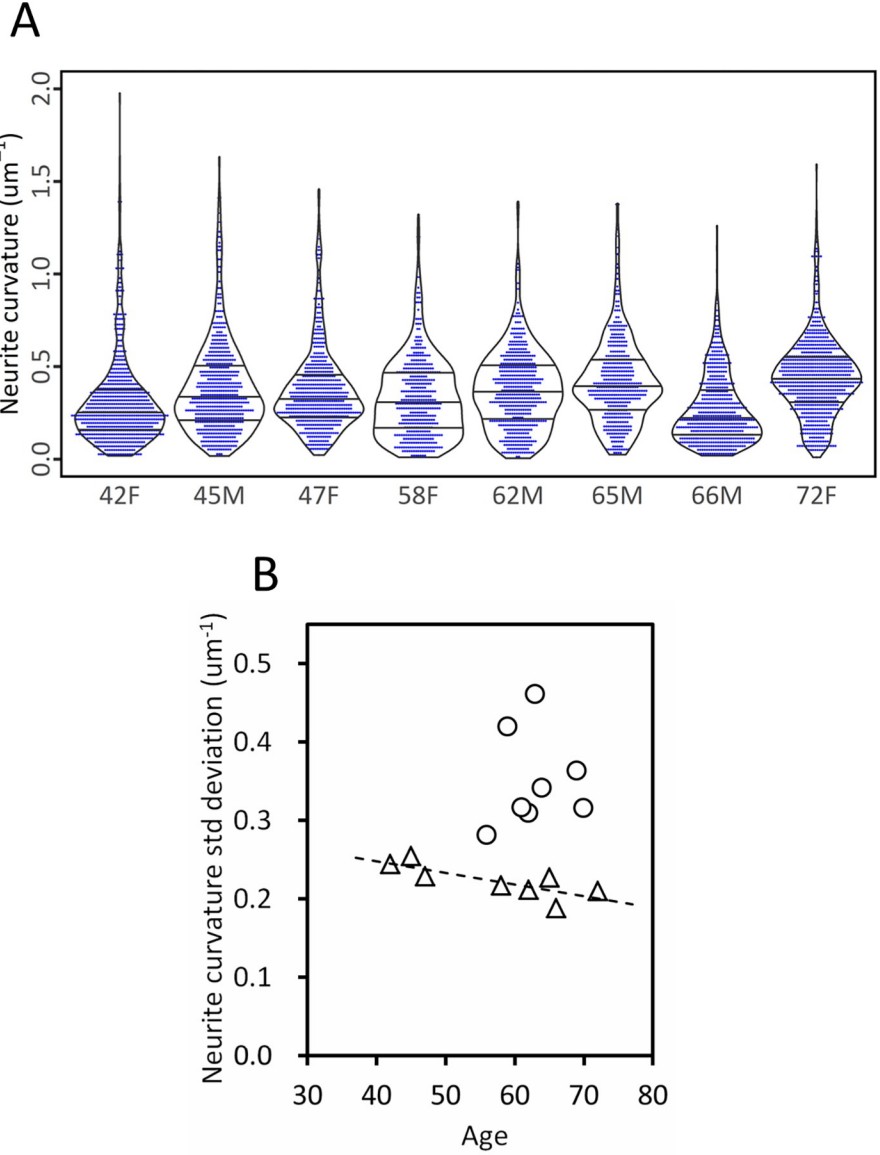

**Fig 2. Relation between neurite curvature and age.** (**A**) Bee-swarm and violin plots of neurite curvature of control cases. Each neurite is indicated with a blue dot. Cases are labeled with their age and sex. Quartiles are indicated with bars. Curvature maximum decreases with age, resulting in a decrease in the distribution span due to aging. This age-dependent decrease in the span leads to a correlation in panel **B**. (**B**) Scatter plot of standard deviation of neurite curvature versus age. Control cases are plotted with triangles and schizophrenia cases with circles. The dashed line indicates a linear regression (Pearson's $r$ = -0.80, $p$ = 0.018).

(S5 Fig). This indicates that the structural changes observed in schizophrenia cases cannot be explained solely from the medication.

## Discussion

The standard deviation of the neurite curvature of the control cases showed a significant correlation with their age (Fig 2B). The distribution profiles of the neurite curvature suggested that the curvature distribution changed due to aging from vertically-symmetric profiles to case-specific profiles (Fig 2A). It has been discussed that the volumetric change in brain tissue due to

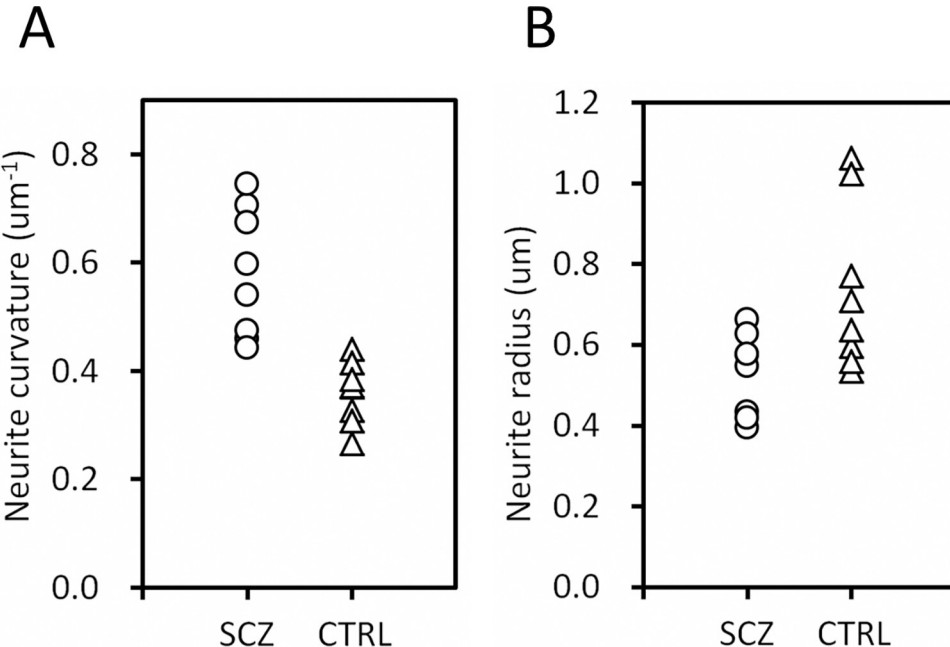

**Fig 3. Differences in structural parameters between schizophrenia and control groups.** (**A**) Neurite curvature ($p = 7.8 \times 10^{-4}$, Welch's $t$-test). (**B**) Neurite radius ($p = 0.028$, Welch's $t$-test).

aging [2, 3] can be ascribed not only to loss of neurons [4–6, 9], but also to morphological changes in neurons [7, 8]. Since most of the electrophysiological properties of neurons, such as resting membrane potential and duration of action potential, remain the same during aging [38], the morphological changes themselves should contribute to the aging of brain function. The age-structure correlation found in this study may be one of the factors affecting aging of the brain. The correlation between the psychiatric score and the structural parameter (Fig 5) suggests that the age-related structural change should have an effect on brain function.

The comparison of the schizophrenia and control groups revealed significant differences in structural parameters (Fig 3). A major difference was observed in the distribution profiles of neurite curvature (Fig 4), suggesting that neurons suffered structural changes from the disorder. The thickness of neurites also showed a significant difference between the schizophrenia and control groups, resulting in thinner neurites in schizophrenia (Fig 3B). According to cable theory [39], neurite thinning should hinder the transmission of active potentials especially between distal neurons. A simulation study using an artificial neural network [40] indicated that a moderate suppression of distal connections improves network performance, though excess suppression degrades network function. We suggest that the tortuous and thin neurites observed in the schizophrenia cases affected the performance of their cortical network of the anterior cingulate cortex that exerts emotional and cognitive functions [41, 42].

Schizophrenia has historically been considered to be a progressive brain disorder according to Kraepelin's definition of *dementia praecox* at the end of 19[th] century [43]. However, longitudinal studies of clinical outcomes and MRI brain volumes have indicated that schizophrenia cannot be considered to be solely a progressive disorder [44]. In this study, we found that the age-related change in the control cases is opposite to the alteration in the schizophrenia cases, of which neurite curvature showed no obvious age dependence (Fig 2B). This indicates that the neuronal alteration in schizophrenia is different from that which occurs during normal aging. Indeed, the analysis of gene expression profiles showed that the expression trajectory

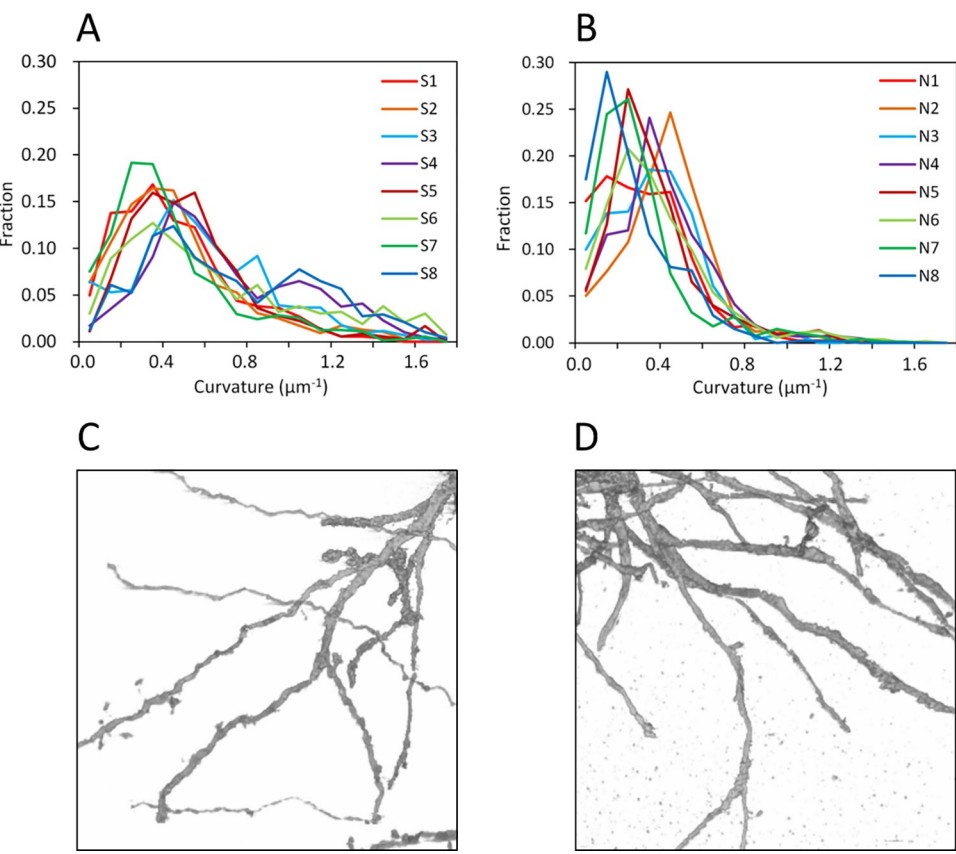

**Fig 4. Differences in neurite structure between schizophrenia and control cases.** (**A**) Frequency distribution of neurite curvatures of schizophrenia cases. Relative frequency in each 0.1 μm$^{-1}$ bin of curvature is plotted. Cases are color-coded. The distribution shows a long tail for every schizophrenia case. (**B**) Frequency distribution of neurite curvature of control cases. The distribution beyond 0.8 μm$^{-1}$ is negligible for every control case. (**C**) Three-dimensional rendering of neurites of the schizophrenia S8A structure. Voxel values 100–800 were rendered with the scatter HQ algorithm of VG Studio Max. Image width: 35 μm. (**D**) Rendering of neurites of the control N5A structure at the same scale.

during aging of schizophrenia cases is different from the age-related change of controls [45]. Although mild cognitive declines in schizophrenia are observed until late adulthood, whether the cognitive decline exceeds those of normal aging remains to be investigated [46]. It is considered that cognitive deficits have already progressed before the prodromal phases [47]. We suggest that the idea of schizophrenia being a progressive disorder should be re-examined by taking account of recent findings.

The schizophrenia S8 case carrying a chromosome 22q11.2 deletion [25] showed the highest neurite curvature (0.74 μm$^{-1}$) among all the cases analyzed in this study (S2 Table). The second highest curvature (0.71 μm$^{-1}$) was observed in the S4 case carrying a *GLO1* mutation [48, 49]. These mutations should be one of the factors that altered the neuronal structures. The 22q11.2 deletion is relevant to neurological disorders such as epilepsy and movement disorders and increases the risk of schizophrenia [50]. Radiographic studies of the 22q11.2 deletion syndrome revealed brain malformations [51], suggesting that this syndrome accompanies histological changes in brain tissue. The *GLO1* gene has also been identified as a locus relevant to schizophrenia [48, 52, 53]. Genetic defects in *GLO1* increase the risk of carbonyl stress [53], which induces damage on neurons [54]. Although genome-wide studies of psychiatric disorders identified a number of disorder-related genes [55], underlying mechanisms linking the

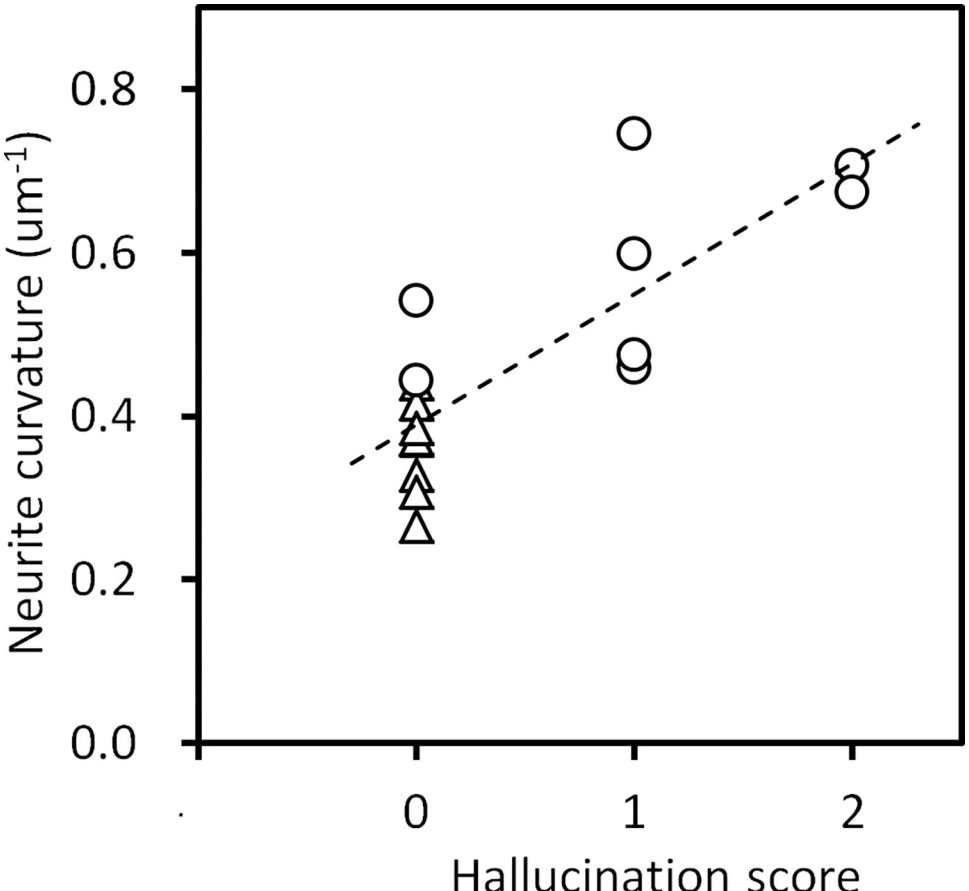

**Fig 5. Scatter plot of neurite curvature and auditory hallucination score.** Schizophrenia cases are plotted with circles and controls with triangles. The dashed line indicates a linear regression (Pearson's $r = 0.80$, $p = 1.8 \times 10^{-4}$).

mutations to the disorders have not been delineated. The two highest curvatures of the mutation cases of this study indicate that neuronal structure should be examined also for other mutations relevant to schizophrenia. We suggest that neuronal structural change plays a pivotal role in linking susceptible genes and disorder symptoms.

It has been reported that the anterior cingulate cortex shows a grey matter reduction in schizophrenia [56, 57] and is associated with auditory hallucinations in schizophrenia [58, 59]. Therefore, brain tissue alteration related to the auditory hallucination should reside in the anterior cingulate cortex of schizophrenia sufferers. In this study, the neurite curvature of this brain area showed a correlation with auditory hallucination score (Fig 5). This result suggests that the hallucination can be reduced by decreasing neurite curvature in the anterior cingulate cortex. Since the neurite curvature can be objectively determined even in animal models, this structural parameter should be used as a hallucination measure with which to exploit antipsychotics or other interventions that can moderate hallucinations. Structural analysis of human brain neurons in other areas will further reveal the relation between the structure and function of neuronal networks and hence will provide information beneficial to sufferers of psychiatric disorders.

## Limitations

There still remains a possibility that the structural changes to neurons observed in schizophrenia cases are caused by antipsychotics, though no obvious correlation was observed between

structural parameters and antipsychotic dose (S5 Fig). It is possible that the cause of death in each case (S2 Table) may have influenced the results. Another limitation of this study is that the analyzed cases are small groups composed of Japanese sufferers. Further analysis of a larger cohort having other racial backgrounds should be conducted to generalize the findings of this study.

## Conclusion

The neurite curvature of the control cases showed a trend depending on age, indicating that the structure of brain neurons changes during aging. Since the structural parameter showed a significant correlation with the psychiatric score, the age-related change should affect the brain function. The neurite curvature of the schizophrenia cases showed long tails in their distribution profiles, indicating that their neurites are thin and tortuous. These structural characteristics of the neurons of the schizophrenia cases should be a possible target of treatments to cure the disorder.

## Supporting information

**S1 Fig.** A–N. Rendering of three-dimensional images of tissue structures and their Cartesian coordinate models. Renderings and models are viewed from nearly the same direction. The pial surface is toward the top. Three-dimensional images were rendered with the scatter HQ algorithm of the VG Studio software. Models were drawn with the MCTrace software. Model constituents are color-coded. Nodes composing each constituent are indicated with octagons. Dots indicate somata nodes. Scale bars: 10 μm.
(PDF)

**S2 Fig.** A–T. Cartesian coordinate models of schizophrenia case structures. The pial surface is toward the top. Models were drawn with the MCTrace software. Constituents of the models are color-coded. Nodes composing each constituent are indicated with octagons. Dots indicate somata nodes. Scale bars: 10 μm.
(PDF)

**S3 Fig.** A–Z. Cartesian coordinate models of control case structures. The pial surface is toward the top. The models were drawn with the MCTrace software. Constituents of the models are color-coded. Nodes composing each constituent are indicated with octagons. Dots indicate somata nodes. Scale bars: 10 μm.
(PDF)

**S4 Fig. Overall three-dimensional structure of S6 and N5 samples.** Layer V is indicated with a box and is colored red. (**A**) Overall structure of the S6 sample. Linear attenuation coefficients of 25–100 cm$^{-1}$ were rendered with the scatter HQ algorithm using the VG Studio software. Image height: 1900 μm. (**B**) Overall structure of the N5 sample. Linear attenuation coefficients of 10–100 cm$^{-1}$ are rendered. Image height: 3520 μm.
(PDF)

**S5 Fig. Scatter plot of neurite curvature and chlorpromazine equivalent dose.** Schizophrenia cases are plotted with circles and controls with triangles.
(PDF)

**S1 Table. Statistics of structural analysis.**
(PDF)

**S2 Table. Structural analysis summary.**
(PDF)

**S3 Table. Conditions of microtomography and nanotomography experiments.**
(PDF)

**S4 Table. Statistics of datasets and Cartesian coordinate models of each case.** (**A–D**)
Schizophrenia cases. (**E–H**) Control cases.
(PDF)

**S1 Data. Individual data of neurite curvature values.**
(XLSX)

## Acknowledgments

We thank Prof. Motoki Osawa and Akio Tsuboi (Tokai University School of Medicine) for their generous support of this study. We also thank the Technical Service Coordination Office of Tokai University for assistance in preparing sample adapters for nanotomography. The synchrotron radiation experiments at SPring-8 were performed with the approval of the Japan Synchrotron Radiation Research Institute (JASRI) (proposal nos. 2011A0034, 2014A1057, 2015A1160, 2019B1087, 2020A0614, 2020A1163, and 2021A1175). The synchrotron radiation experiment at the Advanced Photon Source of Argonne National Laboratory was performed under General User Proposal GUP-59766 and under the approval of the Institutional Biosafety Committee of Argonne National Laboratory. This research used resources of the Advanced Photon Source, a U.S. Department of Energy (DOE) Office of Science User Facility operated for the DOE Office of Science by Argonne National Laboratory under Contract No. DE-AC02-06CH11357.

## Author Contributions

**Conceptualization:** Ryuta Mizutani.

**Data curation:** Ryuta Mizutani, Rino Saiga, Masanari Itokawa.

**Formal analysis:** Ryuta Mizutani, Rino Saiga, Yoshiro Yamamoto.

**Funding acquisition:** Youta Torii, Itaru Kushima, Shuji Iritani, Norio Ozaki.

**Investigation:** Ryuta Mizutani, Rino Saiga, Masayuki Uesugi, Akihisa Takeuchi, Kentaro Uesugi, Yasuko Terada, Yoshio Suzuki, Vincent De Andrade, Francesco De Carlo.

**Methodology:** Ryuta Mizutani, Rino Saiga, Masayuki Uesugi, Akihisa Takeuchi, Kentaro Uesugi, Yasuko Terada, Yoshio Suzuki, Vincent De Andrade, Francesco De Carlo.

**Project administration:** Ryuta Mizutani, Masanari Itokawa.

**Resources:** Ryuta Mizutani, Susumu Takekoshi, Chie Inomoto, Naoya Nakamura, Youta Torii, Itaru Kushima, Shuji Iritani, Norio Ozaki, Kenichi Oshima, Masanari Itokawa, Makoto Arai.

**Software:** Ryuta Mizutani.

**Supervision:** Ryuta Mizutani, Masanari Itokawa.

**Validation:** Ryuta Mizutani, Rino Saiga, Yoshiro Yamamoto.

**Visualization:** Ryuta Mizutani, Rino Saiga.

**Writing – original draft:** Ryuta Mizutani.

**Writing – review & editing:** Ryuta Mizutani, Rino Saiga, Yoshiro Yamamoto, Masayuki Uesugi, Akihisa Takeuchi, Kentaro Uesugi, Yasuko Terada, Yoshio Suzuki, Vincent De Andrade, Francesco De Carlo, Susumu Takekoshi, Chie Inomoto, Naoya Nakamura, Youta Torii, Itaru Kushima, Shuji Iritani, Norio Ozaki, Kenichi Oshima, Masanari Itokawa, Makoto Arai.

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
