## [Decision Letter · Decision Letter 0]

9 May 2023

PONE-D-23-07288Structural aging of human neurons is opposite of the changes in schizophreniaPLOS ONE

Dear Dr. Mizutani,

Thank you for submitting your manuscript to PLOS ONE. After careful consideration, we feel that it has merit but does not fully meet PLOS ONE’s publication criteria as it currently stands. Therefore, we invite you to submit a revised version of the manuscript that addresses the points raised during the review process.

We look forward to receiving your revised manuscript.

Kind regards,

Thiago P. Fernandes, PhD

Academic Editor

PLOS ONE

2. We note that you have a patent relating to material pertinent to this article. Please provide an amended statement of Competing Interests to declare this patent (with details including name and number), along with any other relevant declarations relating to employment, consultancy, patents, products in development or modified products etc. Please confirm that this does not alter your adherence to all PLOS ONE policies on sharing data and materials, as detailed online in our guide for authors http://journals.plos.org/plosone/s/competing-interests by including the following statement: "This does not alter our adherence to  PLOS ONE policies on sharing data and materials.” If there are restrictions on sharing of data and/or materials, please state these. Please note that we cannot proceed with consideration of your article until this information has been declared.

Reviewers' comments:

Reviewer's Responses to Questions

**Comments to the Author**

1. Is the manuscript technically sound, and do the data support the conclusions?

Reviewer #1: Yes

Reviewer #2: Partly

2. Has the statistical analysis been performed appropriately and rigorously? 

Reviewer #1: Yes

Reviewer #2: No

3. Have the authors made all data underlying the findings in their manuscript fully available?

Reviewer #1: Yes

Reviewer #2: No

4. Is the manuscript presented in an intelligible fashion and written in standard English?

Reviewer #1: Yes

Reviewer #2: Yes

5. Review Comments to the Author

Reviewer #1: The referee has followed the nanoCT papers of brain tissue from this group for some time. The data in the present manuscript represent a very large amount of additional work. The referee recommends that the present paper be accepted after so small improvements are made.

The referee is not a brain researcher and cannot comment, therefore, on the literature cited in the Introduction and Discussion. The x-ray imaging will absolutely solid, in the referee’s opinion.

The Results explain the data concisely but clearly. The details are given in the Supplementary Tables, and all of the samples appear in the main figures or the supplementary figures. Thus, reporting is thorough.

In the Discussion, the authors write (lines 335-339) “Although a methodology to manipulate neuron structure of a specific brain area is currently unavailable, neurite curvature can be used as a hallmark with which to exploit antipsychotics or other interventions that can moderate hallucinations.” This is clumsy, in the referee’s opinion, because the first clause says we cannot change neuron structure and the second part of the sentence sounds like neurite curvature is a hallmark with which to exploit antipsychotics…, which cannot be correct because this brain tissue is only available post mortem. The referee is not at all sure what is meant, and this portion of the Discussion must be clear.

The referee was unclear about the designations: “S7A, S7B, S7C…”. This needs to explicitly clarified. Designators S and N clearly refer to diseased and normal tissue. The 7 must be individual 7, one guesses. Does the last letter indicate different portions of tissue from the individual (here #7)?

Minor point: S2 Table lists pixel sizes. One presumes these are actually the reconstructed volume element (voxel) sizes. If so, this should be changed because pixels are 2D features.

Reviewer #2: Dear Authors,

Thanks for your work.

There are a few comments about this work.

1. The introduction does not contain a clear statement of the purpose of the study. The introduction contains an indication of the results that you are yet planning to present here.

2. The description of the norm group is insufficient. There is no indication of what diseases the subjects suffered. It is not excluded the influence of these diseases on the brain.

3. It is necessary to justify the choice of t-test for statistical analysis.

4. In the Discussion section, the wording is questionable: "Cognitive functions in schizophrenia do not worsen over time." The authors provide references to older publications.

5. You need to add a Conclusions section.

6. It is necessary to transfer the information from the Discussion section to the special section "Restrictions of the study".

6. PLOS authors have the option to publish the peer review history of their article (what does this mean?). If published, this will include your full peer review and any attached files.

Reviewer #1: **Yes: **Stuart R. Stock

Reviewer #2: No

---

## [Author Response · Author response to Decision Letter 0]

24 May 2023

PONE-D-23-07288

Point-by-point responses to the reviewer comments

We would like to thank the editor and reviewers for their careful review of our manuscript and for their feedback. We believe this updated manuscript addresses the concerns. The Supporting Information section was moved to the end of the manuscript according to the journal guideline.

Reviewer #1: The referee has followed the nanoCT papers of brain tissue from this group for some time. The data in the present manuscript represent a very large amount of additional work. The referee recommends that the present paper be accepted after so small improvements are made.

The referee is not a brain researcher and cannot comment, therefore, on the literature cited in the Introduction and Discussion. The x-ray imaging will absolutely solid, in the referee’s opinion.

The Results explain the data concisely but clearly. The details are given in the Supplementary Tables, and all of the samples appear in the main figures or the supplementary figures. Thus, reporting is thorough.

We sincerely thank you for your appropriate and fair evaluation of this study. 

In the Discussion, the authors write (lines 335-339) “Although a methodology to manipulate neuron structure of a specific brain area is currently unavailable, neurite curvature can be used as a hallmark with which to exploit antipsychotics or other interventions that can moderate hallucinations.” This is clumsy, in the referee’s opinion, because the first clause says we cannot change neuron structure and the second part of the sentence sounds like neurite curvature is a hallmark with which to exploit antipsychotics…, which cannot be correct because this brain tissue is only available post mortem. The referee is not at all sure what is meant, and this portion of the Discussion must be clear.

Thank you for the suggestion. Since the neurite curvature showed a significant correlation with the auditory hallucination score (Fig. 5), it can be used as a measure of hallucination even in animal disease models. The sentence was revised in order to clarify the discussion (p. 15, lines 345-347 in the file w/o track changes). 

The referee was unclear about the designations: “S7A, S7B, S7C…”. This needs to explicitly clarified. Designators S and N clearly refer to diseased and normal tissue. The 7 must be individual 7, one guesses. Does the last letter indicate different portions of tissue from the individual (here #7)?

Thank you for the suggestion. Dataset S7A stands for dataset A of schizophrenia case S7. This is now explicitly stated in the revised manuscript (p. 6, line 145). Multiple datasets were taken from different portions of the tissue during the same beamtime, or from different tissue samples if the data collection was performed on different beamtimes. The cortical depth of each dataset is listed in S4 Tables A-H. 

Minor point: S2 Table lists pixel sizes. One presumes these are actually the reconstructed volume element (voxel) sizes. If so, this should be changed because pixels are 2D features.

Thank you for the suggestion. The "Image size (pixel)" fields of S4 Tables are now revised as "Image size (voxel)". S3 Table (formerly S2) shows the conditions of the micro/nano-CT experiments; hence, the parameters represent features of 2D pixels. 

Reviewer #2: Dear Authors,

Thanks for your work.

There are a few comments about this work.

1. The introduction does not contain a clear statement of the purpose of the study. The introduction contains an indication of the results that you are yet planning to present here.

Thank you for the suggestion. The sentence which contains an indication of the results was revised in order to state the purpose of this study (p. 3, lines 74-77 in the file w/o track changes). 

2. The description of the norm group is insufficient. There is no indication of what diseases the subjects suffered. It is not excluded the influence of these diseases on the brain. 

Thank you for the suggestion. Causes of death of all the cases are now listed in S2 Table. Cases with diseases known to be directly related to the brain area under analysis were not included in this study. Histological assessment of the cerebral tissues showed no hemorrhage, infarction, or neoplasm. This is stated in p. 5, lines 104-105. It is possible that the cause of death in each case may have influenced the results. This is now stated in the Limitations section (p. 15, lines 355-356). The table numbering was revised in order to cite S2 Table in the first subsection of the Methods. 

3. It is necessary to justify the choice of t-test for statistical analysis.

Since the mean follows a normal distribution, the use of a t-test is justified for the statistical analysis of mean values. The related sentence was revised (p. 7, lines 161-162). Since Shapiro-Wilk normality tests of standard deviations of the neurite curvature showed no statistical significance (p = 0.30 for schizophrenia cases and p = 0.94 for controls), the use of a t-test was justified for the statistical analysis of their difference (p. 7, lines 163-166). The statistical analyses in this study were conducted under the direction of coauthor Yoshiro Yamamoto, whose expertise is statistics. All data needed for the statistical analyses are provided in S1-S4 Tables. 

4. In the Discussion section, the wording is questionable: "Cognitive functions in schizophrenia do not worsen over time." The authors provide references to older publications.

Thank you for the suggestion. The related sentences were revised (p. 14, lines 317-320). The former references were replaced with the following papers: 

46. Fett AJ, Reichenberg A, Velthorst E. Lifespan evolution of neurocognitive impairment in schizophrenia - A narrative review. Schizophr Res Cogn. 2022; 28: 100237.

47. Bora E, Murray RM. Meta-analysis of cognitive deficits in ultra-high risk to psychosis and first-episode psychosis: do the cognitive deficits progress over, or after, the onset of psychosis? Schizophr Bull. 2014; 40: 744–755.

5. You need to add a Conclusions section.

Thank you for the suggestion. The section was added as suggested (p. 16, line 361-368). 

6. It is necessary to transfer the information from the Discussion section to the special section "Restrictions of the study".

The paragraph was removed from the Discussion section to the newly-created "Limitations" section (p. 15, line 352-359). 

Data availability

The S1 Data file is now included in the revised submission to provide individual curvature values. This item is now cited in p. 8, line 185. We are not certain whether this type of data is appropriate for the publication in this journal. If it's appropriate, we would like to include it as Supporting Information. If not, we do not claim its publication. It can be used only for reviewing purposes.

---

## [Decision Letter · Decision Letter 1]

12 Jun 2023

Structural aging of human neurons is opposite of the changes in schizophrenia

PONE-D-23-07288R1

Dear Dr. Mizutani,

Hi - I’d just ask the authors to (1) change “opposite” to another synonym and (2) refine eligibility criteria, like the use of meds, BDNF and other factors that weren’t assessed etc.

We’re pleased to inform you that your manuscript has been judged scientifically suitable for publication and will be formally accepted for publication once it meets all outstanding technical requirements.

Kind regards,

Thiago P. Fernandes, PhD

Academic Editor

PLOS ONE

Additional Editor Comments (optional):

Thank you for your thoughtful and careful edits.

Wishing you success with the study.

Reviewers' comments:

Reviewer's Responses to Questions

**Comments to the Author**

1. If the authors have adequately addressed your comments raised in a previous round of review and you feel that this manuscript is now acceptable for publication, you may indicate that here to bypass the “Comments to the Author” section, enter your conflict of interest statement in the “Confidential to Editor” section, and submit your "Accept" recommendation.

Reviewer #1: All comments have been addressed

Reviewer #2: All comments have been addressed

2. Is the manuscript technically sound, and do the data support the conclusions?

Reviewer #1: Yes

Reviewer #2: Yes

3. Has the statistical analysis been performed appropriately and rigorously? 

Reviewer #1: I Don't Know

Reviewer #2: Yes

4. Have the authors made all data underlying the findings in their manuscript fully available?

Reviewer #1: Yes

Reviewer #2: (No Response)

5. Is the manuscript presented in an intelligible fashion and written in standard English?

Reviewer #1: Yes

Reviewer #2: Yes

6. Review Comments to the Author

Reviewer #1: Accept. The authors have addressed the points this referee raised. They also seem to have covered the points raised by the other referee.

Reviewer #2: (No Response)

7. PLOS authors have the option to publish the peer review history of their article (what does this mean?). If published, this will include your full peer review and any attached files.

Reviewer #1: **Yes: **Stuart R. Stock

Reviewer #2: No

---

## [Editor Report · Acceptance letter]

16 Jun 2023

PONE-D-23-07288R1 

Structural aging of human neurons is opposite of the changes in schizophrenia 

Dear Dr. Mizutani:

I'm pleased to inform you that your manuscript has been deemed suitable for publication in PLOS ONE. Congratulations! Your manuscript is now with our production department. 

Kind regards, 

on behalf of

Dr. Thiago P. Fernandes 

Academic Editor

PLOS ONE